# Molecular and Cellular Mechanisms of Immunosenescence: Modulation Through Interventions and Lifestyle Changes

**DOI:** 10.3390/biology14010017

**Published:** 2024-12-27

**Authors:** Luca Pangrazzi, Andreas Meryk

**Affiliations:** 1Institute for Biomedical Aging Research, Faculty of Biology, University of Innsbruck, 6020 Innsbruck, Austria; luca.pangrazzi-1@unitn.it; 2Department of Pediatrics, Medical University of Innsbruck, 6020 Innsbruck, Austria

**Keywords:** immunosenescence, inflammaging, oxidative stress, interventions

## Abstract

As we age, our immune system becomes less effective, a process called immunosenescence. This weakening of the immune system makes older adults more susceptible to infections, reduces the effectiveness of vaccines, and increases the risk of age-related diseases. Our review examines the complex biological mechanisms behind immunosenescence and explores how lifestyle factors like diet, exercise, and stress management can influence this process. We discuss recent scientific findings that shed light on how aging affects immune cells at the molecular level and how these changes contribute to chronic low-grade inflammation often seen in older adults. Understanding these processes is crucial for developing strategies to maintain a healthy immune system as we age. Our research suggests that adopting certain lifestyle habits and potential medical interventions could help slow down or partially reverse some aspects of immune aging, potentially leading to better health outcomes for older populations.

## 1. Introduction

The global population is aging at an unprecedented rate. According to the World Health Organization (WHO), the number of people aged 60 years and older is expected to double by 2050, reaching approximately 2 billion. This demographic shift poses significant challenges for healthcare systems worldwide, as older adults are more prone to chronic diseases such as cardiovascular diseases, diabetes, cancer, and neurodegenerative disorders [1,2,3].

### 1.1. Inflammaging and Immunosenescence

Inflammaging refers to a state of chronic, low-grade inflammation that develops with advancing age. This phenomenon is characterized by increased levels of pro-inflammatory cytokines and acute-phase proteins in the absence of overt infection. Inflammaging is thought to contribute to the pathogenesis of various age-related diseases and is considered a hallmark of the aging process [4,5,6,7].

The concept of inflammaging was first introduced by Franceschi et al. in 2000, and since then, it has gained significant attention in the field of gerontology. Key features of inflammaging include elevated levels of inflammatory markers such as C-reactive protein (CRP), interleukin-6 (IL-6), and tumor necrosis factor-alpha (TNF-α). These chronic inflammatory states can lead to tissue damage, functional decline, and increased vulnerability to age-related diseases [8,9].

Immunosenescence, on the other hand, describes the gradual deterioration of the immune system associated with aging. This decline affects both the innate and adaptive branches of the immune system, leading to increased susceptibility to infections, reduced vaccine efficacy, and a higher incidence of autoimmune diseases and cancer. Immunosenescence is characterized by several key changes [4,6,8,10,11] as follows:Decreased production of naive T cells due to thymic involutionAccumulation of memory T cells with reduced diversity and functionalityImpaired B cell function and antibody productionReduced efficacy of innate immune cells such as neutrophils and natural killer cells

The interplay between immunosenescence and inflammaging creates a vicious cycle that exacerbates the aging process and contributes to the development of age-related pathologies. For instance, senescent cells accumulate with age and adopt a senescence-associated secretory phenotype (SASP), which contributes to the pro-inflammatory environment characteristic of inflammaging [12,13].

Understanding the mechanisms underlying inflammaging and immunosenescence is crucial for developing interventions to promote healthy aging and reduce the burden of age-related diseases. Current research focuses on strategies to modulate these processes, including lifestyle interventions, nutritional approaches, and pharmacological treatments.

### 1.2. Impact on Health and Disease

The interplay between inflammaging and immunosenescence has profound implications for health and disease in the elderly. Chronic inflammation and immune dysregulation contribute to the development and progression of various age-related diseases, including cardiovascular diseases, type 2 diabetes, Alzheimer’s disease, and certain cancers. Understanding the mechanisms driving these processes is crucial for developing targeted interventions to promote healthy aging and improve the quality of life for older adults [14,15].

In cardiovascular diseases, the effects of immunosenescence and inflammaging are particularly notable. Chronic inflammation leads to increased levels of proinflammatory cytokines, resulting in endothelial injury, vascular remodeling damage, and accelerated atherosclerosis. The inflammaging condition recruits monocytes and triggers their conversion into lipid-containing, senescent foamy macrophages, further exacerbating atherosclerotic progression. Moreover, the expansion of CD8 + CD28- T-cells, especially in cytomegalovirus-infected patients, has been identified as a risk factor for vascular dysfunction and is strongly associated with atherosclerosis and acute coronary syndrome [14]

Neurodegenerative diseases, particularly Alzheimer’s disease (AD), are also significantly impacted by immunosenescence. AD patients typically exhibit lower levels of naive T cells, higher levels of memory T cells, and significant telomere shortening in T cells. Senescent T cells enhance the proinflammatory effects of microglia, promoting neuroinflammation. Additionally, the accumulation of regulatory T cells (Tregs) in the peripheral immune system can impair the infiltration of inflammation-resolving immune cells into the central nervous system, potentially exacerbating neurodegenerative processes [16].

The role of immunosenescence in cancer development and progression is equally significant. The aging immune system shows a reduced ability to recognize and eliminate malignant cells, while impaired tumor-specific T cell responses contribute to an increased cancer risk in the elderly. Furthermore, the accumulation of senescent cells promotes a pro-tumorigenic microenvironment, potentially facilitating cancer growth and metastasis [15,17].

Metabolic diseases are also influenced by the chronic inflammation associated with immunosenescence. Inflammaging is implicated in the development of type 2 diabetes and obesity, with senescent immune cells exhibiting a proinflammatory senescence-associated secretory phenotype (SASP) that exacerbates metabolic dysfunction [5].

Immunosenescence leads to increased vulnerability to infectious diseases due to reduced efficacy of innate immune responses, including decreased neutrophil function and impaired natural killer cell cytotoxicity. The diminished adaptive immune responses, particularly in T cell-mediated immunity, and compromised barrier functions of the skin and mucous membranes further contribute to this susceptibility.

Another critical consequence of immunosenescence is the reduced efficacy of vac-cines in older populations. The age-related decline in immune function results in lower antibody titers following vaccination, reduced duration of protective immunity, and a decreased ability to respond to new antigens. This diminished vaccine effectiveness poses significant challenges in protecting the elderly population against various infectious diseases [15,18,19,20].

### 1.3. Influence of Sex on Inflammaging and Immunosenescence

Sex plays a critical role in shaping the aging process of the immune system, influencing both inflammaging and immunosenescence. Significant differences in immune responses and aging patterns between males and females have been observed, largely attributed to sex hormones, genetic factors, and environmental influences [21,22,23]. Estrogen, the primary female sex hormone, generally exhibits anti-inflammatory properties and enhances immune responses. In contrast, testosterone, the main male sex hormone, typically suppresses immune function [24]. These hormonal differences contribute to the sexual dimorphism observed in immune aging as follows:Inflammaging: Women tend to have higher levels of pro-inflammatory markers associated with inflammaging, such as C-reactive protein (CRP) and interleukin-6 (IL-6), compared to men of the same age. This heightened inflammatory state in females may contribute to their increased susceptibility to certain autoimmune diseases [25,26].T cell responses: Females generally maintain higher numbers of CD4+ T cells and exhibit stronger T cell responses throughout life compared to males. This may contribute to more robust vaccine responses in elderly women but also increases their risk of T cell-mediated autoimmune diseases [25].B cell function: Women tend to produce higher levels of antibodies and maintain better B cell function with age compared to men. This may explain the higher efficacy of certain vaccines in elderly females [27].Natural Killer (NK) cells: Males typically have higher numbers and activity of NK cells, which may contribute to their lower incidence of certain autoimmune diseases but potentially increased susceptibility to viral infections [28].Thymic involution: The thymus, crucial for T cell development, undergoes age-related involution at different rates between sexes. Males experience more rapid thymic involution, potentially contributing to faster decline in naive T cell production [29].Telomere attrition: Females generally maintain longer telomeres throughout life, which may contribute to their longer life expectancy and potentially slower rate of immunosenescence [30].Epigenetic changes: Sex-specific epigenetic alterations during aging affect immune gene expression differently in males and females, contributing to divergent immunosenescence patterns [31].

Understanding these sex-based differences in immune aging is crucial for developing targeted interventions and personalized approaches to healthy aging. Future research should focus on elucidating the mechanisms underlying these differences and exploring sex-specific strategies to mitigate immunosenescence and inflammaging [32].

## 2. Molecular Mechanisms of Immunosenescence

### 2.1. Telomere Attrition

One of the primary molecular mechanisms underlying immunosenescence is telomere attrition. Telomeres, the repetitive DNA sequences at the ends of chromosomes, shorten with each cell division. In immune cells, particularly T lymphocytes, this shortening leads to replicative senescence [33,34]. Studies have shown that telomere length in lymphocytes is a biomarker of immune system aging, with shorter telomeres associated with reduced proliferative capacity and altered cytokine production [34]. The enzyme telomerase, responsible for maintaining telomere length, shows reduced activity in aged immune cells. This decrease in telomerase activity contributes to the accumulation of senescent T cells, characterized by the loss of CD28 expression. These CD28- T cells exhibit a senescence-associated secretory phenotype (SASP), producing pro-inflammatory cytokines that contribute to chronic inflammation [35].

### 2.2. Epigenetic Alterations

Epigenetic changes play a crucial role in immunosenescence. Age-related alterations in DNA methylation patterns and histone modifications affect gene expression in immune cells, leading to functional impairments [36]. For instance, changes in the methylation status of genes involved in T cell differentiation and function, such as IFN-γ and IL-7R, have been observed in aged T cells [37].

Histone modifications, particularly the loss of repressive marks like H3K27me3, contribute to the aberrant expression of inflammatory genes in senescent immune cells. These epigenetic alterations not only affect cellular function but also contribute to the heterogeneity of immune responses observed in older individuals [38,39].

### 2.3. DNA Damage, Genomic Instability and Mitochondrial Dysfunction

Accumulation of DNA damage and genomic instability are hallmarks of aging that significantly impact immune function. Persistent DNA damage activates the DNA damage response (DDR) pathway, leading to cell cycle arrest and senescence. In immune cells, this results in a reduced pool of functional cells capable of responding to new antigens. Moreover, genomic instability can lead to chromosomal abnormalities and altered gene expression patterns in immune cells [40,41]. For instance, studies have shown an increased frequency of chromosomal translocations in aged B cells, potentially contributing to the higher incidence of B cell malignancies in older adults [42].

Mitochondrial dysfunction is another key factor in immunosenescence. Aging is associated with an accumulation of mitochondrial DNA mutations and a decline in mitochondrial function. In immune cells, particularly T lymphocytes, this leads to reduced energy production and increased oxidative stress [43,44]. The impaired mitochondrial function affects various aspects of T cell biology, including activation, proliferation, and memory formation. For example, aged naive T cells show reduced mitochondrial mass and decreased spare respiratory capacity, limiting their ability to undergo clonal expansion upon antigen stimulation [45,46,47].

## 3. Signaling Pathways in Immunosenescence

The molecular changes associated with immunosenescence are reflected in alterations of various signaling pathways crucial for immune cell function and homeostasis.

### 3.1. NF-κB Signaling and mTOR Pathway

The nuclear factor kappa B (NF-κB) pathway plays a central role in immune responses and inflammation. In aging, there is a chronic activation of NF-κB signaling, contributing to the low-grade inflammation characteristic of immunosenescence. This persistent NF-κB activation leads to increased production of pro-inflammatory cytokines and chemokines, further exacerbating the inflammatory state [48,49].

Interestingly, the NF-κb pathway also interacts with other age-related pathways, such as the mammalian target of rapamycin (mTOR) signaling, creating a complex network that regulates cellular senescence and inflammation. The mTOR pathway, a key regulator of cellular metabolism and growth, is dysregulated in aging immune cells. Persistent mTOR activation has been linked to impaired autophagy, accumulation of damaged cellular components, and reduced lifespan [50].

In T cells, hyperactivation of mTOR signaling contributes to the loss of CD28 expression and the acquisition of a senescent phenotype. Moreover, mTOR activation skews T cell differentiation towards short-lived effector cells at the expense of memory cell formation, potentially explaining the reduced immunological memory in older individuals [51,52].

### 3.2. AMPK Signaling

AMP-activated protein kinase (AMPK) is a crucial energy sensor that regulates cellular metabolism. In aging immune cells, AMPK activity is often reduced, leading to metabolic dysfunction and impaired responses to stress.

The decline in AMPK signaling affects various aspects of immune function, including T cell memory formation and maintenance. Strategies to activate AMPK, such as metformin treatment, have shown promise in enhancing immune function in aged individuals [53,54].

### 3.3. Sirtuin Pathways

Sirtuins, a family of NAD+-dependent deacetylases, play important roles in regulating cellular stress responses and lifespan. In immune cells, sirtuins modulate various processes, including inflammation, T cell differentiation, and metabolic reprogramming [55].

Age-related decline in sirtuin activity, particularly SIRT1 and SIRT3, contributes to increased inflammation and reduced stress resistance in immune cells. Enhancing sirtuin activity through NAD+ precursors or small molecule activators has emerged as a potential strategy to mitigate aspects of immunosenescence [56,57].

In conclusion, immunosenescence is driven by complex molecular mechanisms and alterations in key signaling pathways. Understanding these processes provides opportunities for developing targeted interventions to maintain immune function in aging populations. Future research should focus on integrating these molecular insights with lifestyle and pharmacological interventions to promote healthy immune aging.

### 3.4. Melatonin Signaling Pathway

Melatonin, a hormone primarily produced by the pineal gland, plays a crucial role in regulating circadian rhythms and has emerged as an important modulator of immunosenescence. The melatonin signaling pathway involves several mechanisms that contribute to its anti-aging effects as follows:

CREB Signaling: Melatonin activates the cAMP response element-binding protein (CREB) signaling pathway, which is associated with long-term memory processing and neuroprotection. This activation occurs through the Raf-ERK-p90RSK pathway, independent of calcium channels, JNK, or Akt signaling [58,59].

Antioxidant Effects: Melatonin acts as a powerful antioxidant by stimulating antioxidant enzymes such as glutathione peroxidase, superoxide dismutase, and catalase. This helps combat oxidative stress associated with aging [60].

Mitochondrial Function: Melatonin improves mitochondrial function by reducing oxidative phosphorylation deficits and increasing ATP production, which is crucial for maintaining cellular energy homeostasis during aging [61,62].

Anti-inflammatory Actions: Melatonin suppresses the production of pro-inflammatory cytokines and modulates the activity of immune cells, contributing to the attenuation of inflammaging [58].

Sirtuin Activation: Melatonin has been shown to activate sirtuins, particularly SIRT1, which are important regulators of cellular stress responses and metabolism [63,64].

### 3.5. Klotho Signaling Pathway

The Klotho gene, named after the Greek goddess who spins the thread of life, is a critical regulator of aging processes. The Klotho protein participates in several signaling pathways that influence immunosenescence [65]:

FGF23 Signaling: Klotho acts as a co-receptor for fibroblast growth factor 23 (FGF23), regulating phosphate homeostasis and vitamin D metabolism. This pathway is crucial for maintaining mineral balance and preventing vascular calcification associated with aging [66].

Insulin/IGF-1 Signaling: Klotho suppresses insulin/IGF-1 signaling, which is associated with extended lifespan. This effect may contribute to the regulation of cellular stress responses and metabolism during aging [67].

Wnt Signaling: Klotho inhibits Wnt signaling, which is involved in stem cell function and tissue regeneration. Modulation of this pathway by Klotho may influence tissue homeostasis during aging [68].

Oxidative Stress Regulation: Klotho enhances resistance to oxidative stress by activating the FOXO forkhead transcription factors, which regulate antioxidant enzymes [69].

Calcium Homeostasis: Klotho regulates calcium homeostasis through its effects on transient receptor potential vanilloid 5 (TRPV5) channels, which may influence cellular senescence and tissue function [70].

## 4. Inflammaging: Molecular Drivers and Consequences

### 4.1. Senescence-Associated Secretory Phenotype (SASP)

One of the primary contributors to inflammaging is the accumulation of senescent cells and their associated secretory phenotype. Cellular senescence is a state of permanent cell cycle arrest that occurs in response to various stressors, including DNA damage, oxidative stress, and telomere attrition. While senescence serves as a tumor suppressor mechanism, senescent cells can have detrimental effects on tissue homeostasis and function through the SASP [13,71].

The SASP is characterized by the secretion of a complex mixture of pro-inflammatory cytokines, chemokines, growth factors, and matrix-degrading enzymes. Key components of the SASP include [72,73] the following:Pro-inflammatory cytokines: IL-6, IL-1α, IL-1β, TNF-αChemokines: IL-8, MCP-1, MIP-1αGrowth factors: VEGF, HGF, TGF-βMatrix metalloproteinases: MMP-1, MMP-3, MMP-10

These SASP factors contribute to chronic inflammation by attracting immune cells, altering the tissue microenvironment, and potentially inducing senescence in neighboring cells through paracrine mechanisms. The accumulation of senescent cells with age leads to a progressive increase in SASP factors, creating a self-perpetuating cycle of inflammation and cellular dysfunction. Recent studies have shown that the SASP is regulated by various signaling pathways, including NF-κB, p38 MAPK, and mTOR. Targeting these pathways or selectively eliminating senescent cells (senolysis) has emerged as a promising strategy to mitigate inflammaging and its associated pathologies [74,75,76].

### 4.2. Chronic Activation of Innate Immune Sensors

Another key driver of inflammaging is the chronic activation of innate immune sensors, particularly pattern recognition receptors (PRRs) such as Toll-like receptors (TLRs) and NOD-like receptors (NLRs). These receptors recognize pathogen-associated molecular patterns (PAMPs) and damage-associated molecular patterns (DAMPs), triggering inflammatory responses [77,78].

With age, there is an increased accumulation of endogenous DAMPs, including the following [11,79,80]:Cell-free DNA and RNAOxidized lipids and proteinsAdvanced glycation end products (AGEs)Extracellular ATP and uric acid crystals

These DAMPs can persistently activate PRRs, leading to chronic low-grade inflammation [81]. For example, cell-free mitochondrial DNA, which increases with age, can activate TLR9 and the NLRP3 inflammasome, promoting the production of pro-inflammatory cytokines. Moreover, age-related changes in the expression and function of PRRs contribute to dysregulated innate immune responses. Studies have shown altered TLR expression and signaling in aged immune cells, potentially leading to impaired pathogen recognition and exaggerated inflammatory responses to endogenous ligands [82,83,84]. The chronic activation of innate immune sensors not only contributes to systemic inflammation but also impacts adaptive immunity. Persistent PRR signaling can lead to T cell exhaustion and impaired B cell responses, further compromising immune function in older individuals [6,12].

### 4.3. Gut Microbiome Dysbiosis and Inflammaging

The gut microbiome plays a crucial role in maintaining immune homeostasis and has emerged as a significant factor in inflammaging. Age-related changes in the composition and function of the gut microbiota, known as dysbiosis, can contribute to chronic inflammation through several mechanisms [85].

Key features of age-related gut dysbiosis include [86,87] the following:Reduced microbial diversityDecreased abundance of beneficial bacteria (e.g., Bifidobacterium)Increased proportion of pro-inflammatory species (e.g., Proteobacteria)Altered microbial metabolite production

These changes can lead to increased intestinal permeability, often referred to as “leaky gut”, allowing the translocation of bacterial products such as lipopolysaccharide (LPS) into the circulation. LPS can activate TLR4 on immune cells, triggering systemic inflammation. Furthermore, age-related dysbiosis affects the production of microbial metabolites that modulate immune function. For instance, short-chain fatty acids (SCFAs), produced by the fermentation of dietary fiber, have anti-inflammatory properties and regulate T cell differentiation. A decrease in SCFA-producing bacteria with age may contribute to inflammaging and impaired immune responses [88,89,90].

Recent studies have also highlighted the role of the gut–brain axis in inflammaging [91,92,93]. Microbial dysbiosis can influence neuroinflammation and cognitive decline through the production of neurotoxic metabolites and the modulation of systemic inflammation.

Interventions targeting the gut microbiome, such as probiotics, prebiotics, and fecal microbiota transplantation, have shown promise in reducing inflammation and improving immune function in older adults. However, more research is needed to fully elucidate the complex interactions between the aging gut microbiome and the immune system [89].

### 4.4. Metabolic Inflammation

Metabolic inflammation, also known as metaflammation, is a key contributor to inflammaging and is closely linked to age-related metabolic disorders such as obesity, type 2 diabetes, and cardiovascular disease [94,95]. This chronic, low-grade inflammation is characterized by the activation of inflammatory pathways in metabolic tissues, particularly adipose tissue [77].

Several factors contribute to metabolic inflammation in aging as follows:

Adipose tissue dysfunction: with age, there is an increase in visceral adiposity and a decline in the function of subcutaneous adipose tissue. Senescent adipocytes and infiltrating immune cells in adipose tissue produce pro-inflammatory cytokines, contributing to systemic inflammation [95,96].

Insulin resistance: age-related insulin resistance leads to hyperglycemia and elevated free fatty acids, which can activate inflammatory pathways through oxidative stress and ER stress mechanisms [97,98].

Mitochondrial dysfunction: aging is associated with a decline in mitochondrial function, leading to increased production of reactive oxygen species (ROS) and activation of inflammatory pathways such as NF-κB [43].

Altered nutrient sensing: dysregulation of nutrient-sensing pathways, including mTOR and AMPK, can lead to impaired autophagy and increased cellular stress, contributing to inflammation [98].

The interplay between metabolic inflammation and immunosenescence creates a vicious cycle that accelerates age-related decline. For example, pro-inflammatory cytokines produced by adipose tissue can impair T cell function and promote the accumulation of senescent immune cells [77].

Interventions targeting metabolic inflammation, such as caloric restriction, exercise, and pharmacological approaches (e.g., metformin), have shown promise in reducing inflammaging and improving healthspan in preclinical models. These strategies aim to restore metabolic homeostasis and reduce the chronic activation of inflammatory pathways [99,100].

In conclusion, inflammaging is driven by multiple interconnected molecular mechanisms, including the SASP, chronic activation of innate immune sensors, gut microbiome dysbiosis, and metabolic inflammation. Understanding these drivers and their consequences is crucial for developing targeted interventions to promote healthy aging and reduce the burden of age-related diseases. Future research should focus on integrating these various aspects of inflammaging to develop comprehensive strategies for maintaining immune health and overall well-being in older adults.

## 5. Emerging Research to Fight Against or at Least Decelerate Immunosenescence

### 5.1. Potential Molecular Targets for Intervention

As our understanding of the molecular mechanisms underlying immunosenescence and inflammaging deepens, several promising targets for intervention have emerged. These targets offer potential avenues for developing therapies to mitigate the effects of age-related immune decline and chronic inflammation (Table 1).

These molecular targets offer promising avenues for developing interventions to combat immunosenescence and inflammaging. However, it is important to note that aging is a complex, multifaceted process, and effective interventions will likely require a combinatorial approach targeting multiple pathways simultaneously. Furthermore, as our understanding of the molecular mechanisms of aging continues to evolve, new targets and intervention strategies are likely to emerge.

### 5.2. Life Style Changes to Escape Immunosenescence

#### 5.2.1. Physical Activity Intervention

Regular physical activity has emerged as one of the most promising interventions to combat immunosenescence and inflammaging. A growing body of evidence suggests that habitual exercise can significantly modulate immune function, potentially slowing the age-related decline in immune competence and reducing chronic low-grade inflammation associated with aging.

Moderate-intensity exercise, performed regularly, has been shown to enhance various aspects of immune function. Studies have demonstrated that physically active older adults exhibit improved neutrophil chemotaxis, increased natural killer cell cytotoxicity, and enhanced T lymphocyte proliferation compared to their sedentary counterparts. These improvements in cellular immunity are thought to contribute to a reduced risk of infectious diseases and potentially a decreased incidence of certain cancers in active individuals [125].

The beneficial effects of exercise on the immune system are believed to be mediated through several mechanisms. One key pathway involves the transient elevation of circulating leukocytes and cytokines that occurs during and immediately after exercise. This “exercise-induced lymphocytosis” is thought to enhance immune surveillance by promoting the exchange of immune cells between the circulation and tissues. Over time, repeated bouts of exercise may lead to a cumulative enhancement of immune function [126].

Moreover, regular physical activity has been associated with lower levels of pro-inflammatory cytokines, such as interleukin-6 (IL-6) and tumor necrosis factor-alpha (TNF-α), which are hallmarks of inflammaging. This anti-inflammatory effect of exercise may help to mitigate the chronic low-grade inflammation that contributes to various age-related diseases and functional decline [127].

The type, intensity, and duration of exercise appear to be important factors in determining its impact on immune function. While moderate-intensity exercise is generally associated with positive immune outcomes, prolonged high-intensity exercise may temporarily suppress certain aspects of immune function, potentially increasing susceptibility to infections in the short term. This has led to the development of the “J-shaped curve” hypothesis, which suggests that moderate exercise reduces infection risk, while excessive exercise may increase it [128].

Recent research has also highlighted the potential of resistance training in modulating immune function in older adults. A systematic review by Sellami et al. found that resistance exercise can improve various immune parameters, including increased lymphocyte proliferation and natural killer cell activity. These findings suggest that a combination of aerobic and resistance exercise may provide optimal benefits for immune health in aging populations [48].

The duration and frequency of exercise required to achieve immune benefits remain subjects of ongoing research. However, current evidence suggests that adhering to public health guidelines for physical activity (150 min of moderate-intensity or 75 min of vigorous-intensity aerobic activity per week, along with muscle-strengthening activities) is likely to confer significant immune benefits [129,130].

Interestingly, the effects of exercise on immune function may be particularly pronounced in older adults. A study by Wong et al. demonstrated that physically active older women had a more robust antibody response to influenza vaccination compared to their inactive peers, suggesting that exercise may enhance vaccine efficacy in this population [131,132].

Furthermore, regular physical activity has been associated with the maintenance of thymic mass and function, which typically declines with age. The thymus gland plays a crucial role in T cell development and maturation, and its preservation may contribute to a more robust and diverse T cell repertoire in physically active older adults [133].

It is important to note that while exercise offers numerous benefits for immune function, it should be approached judiciously, especially in older adults or those with pre-existing health conditions. Gradual progression in exercise intensity and duration, along with proper recovery, is essential to maximize benefits while minimizing risks.

In conclusion, regular physical activity represents a powerful and accessible intervention for maintaining immune function and reducing inflammation in aging populations. As our understanding of the complex interactions between exercise and immunity continues to grow, tailored exercise prescriptions may become an increasingly important tool in promoting healthy aging and resilience against age-related diseases.

#### 5.2.2. Healthy Diet: The Mediterranean Approach to Supporting Immune Health

A healthy diet plays a crucial role in maintaining a robust immune system and reducing chronic inflammation, both of which are key factors in healthy aging. The Mediterranean diet, in particular, has gained significant attention for its potential to support immune function and overall health. This dietary pattern is characterized by high consumption of fruits, vegetables, whole grains, legumes, nuts, and olive oil, moderate consumption of fish and poultry, and limited intake of red meat and processed foods.

Omega-3 Fatty Acids:

Omega-3 fatty acids, particularly eicosapentaenoic acid (EPA) and docosahexaenoic acid (DHA), have potent anti-inflammatory properties. These fatty acids are abundant in fatty fish such as salmon, sardines, and mackerel, which are staples of the Mediterranean diet. Plant-based sources like walnuts, flaxseeds, and chia seeds also provide alpha-linolenic acid (ALA), a precursor to EPA and DHA. Regular consumption of omega-3s has been associated with reduced inflammation and enhanced immune cell function [134,135,136]. Studies have shown that omega-3 fatty acids can enhance immune cell function in several ways. For instance, EPA and DHA have been found to increase phagocytosis by neutrophils and macrophages, enhance natural killer cell activity, and modulate T cell differentiation towards anti-inflammatory phenotypes [134,137,138].

Antioxidants:

The Mediterranean diet is rich in colorful fruits and vegetables, which are excellent sources of antioxidants. These compounds, including vitamins C and E, carotenoids, and polyphenols, help protect cells from oxidative stress and support immune function. For instance, citrus fruits provide vitamin C, while leafy greens offer a variety of antioxidants. The diverse array of plant foods in this diet ensures a broad spectrum of antioxidants, each with unique benefits for immune health [121,139,140]. Vitamin C has been shown to enhance T cell proliferation and differentiation, while vitamin E improves T cell-mediated function and lymphocyte proliferation in older adults [141,142].

Zinc:

While not typically associated with Mediterranean cuisine, zinc-rich foods can be incorporated into this dietary pattern. Oysters, lean beef, and pumpkin seeds are good sources of zinc. This mineral plays a crucial role in the development and function of immune cells. Including moderate amounts of lean meats and seeds in the diet can help ensure adequate zinc intake [143,144,145]. Zinc is crucial for the development and function of neutrophils, natural killer cells, and T lymphocytes. Studies have demonstrated that zinc supplementation can restore thymic function and improve T cell numbers and function in older adults [145,146,147,148].

Vitamin D:

Vitamin D is essential for immune function and can be obtained through sun exposure or dietary sources. Fatty fish, egg yolks, and fortified dairy products can contribute to vitamin D intake. In the Mediterranean region, regular outdoor activities and sun exposure traditionally helped maintain vitamin D levels. However, in modern times, supplementation may be necessary for individuals with limited sun exposure or those at risk of deficiency [149,150,151]. Vitamin D receptors are expressed on various immune cells, including T cells, B cells, and antigen-presenting cells. Research has shown that vitamin D can enhance the antimicrobial effects of macrophages and monocytes, and modulate T cell responses to reduce inflammation [152,153,154].

Probiotics:

Fermented foods, which are natural sources of probiotics, are common in Mediterranean cuisine. Yogurt, kefir, and fermented vegetables like olives and pickles contribute beneficial bacteria to the gut microbiome. These probiotics play a crucial role in maintaining gut health and modulating the immune system. The interaction between gut bacteria and immune cells is a growing area of research, highlighting the importance of a healthy gut microbiome for overall immune function [122,123,124]. Probiotic strains such as Lactobacillus and Bifidobacterium have been shown to enhance natural killer cell activity, improve phagocytosis by neutrophils, and modulate cytokine production, potentially reducing age-related inflammation [155,156,157].

#### 5.2.3. Lifestyle Interventions

Lifestyle interventions play a crucial role in mitigating the effects of immunosenescence and promoting healthy aging. A holistic approach that addresses various aspects of daily life can significantly impact immune function and overall well-being in older adults.

Stress management is a key component of lifestyle interventions targeting immunosenescence. Chronic stress has been shown to accelerate the aging of the immune system, leading to increased inflammation and reduced immune function. Practices such as meditation, yoga, and deep breathing exercises have demonstrated potential in reducing stress and supporting immune function. A study by Black et al. found that mindfulness meditation reduced pro-inflammatory gene expression in older adults, specifically decreasing the activity of NF-κB-related genes, which are key regulators of inflammation [126,158,159,160].

Adequate sleep is another critical factor in maintaining a healthy immune system. Aiming for 7–9 h of quality sleep per night is essential, as sleep deprivation can negatively impact immune function. Research has shown that poor sleep quality and quantity are associated with increased susceptibility to infections and reduced vaccine efficacy in older adults. Implementing good sleep hygiene practices and addressing sleep disorders can help support immune function and overall health [161,162]. Moreover, research has shown that individuals sleeping less than 6 h per night have a 4.2 times higher risk of catching a cold compared to those sleeping more than 7 h, highlighting the importance of adequate sleep for immune function [163,164].

Maintaining strong social connections has emerged as an important factor in healthy aging and immune function. Social isolation and loneliness have been linked to increased inflammation and reduced immune function in older adults. Engaging in social activities, maintaining relationships with family and friends, and participating in community events can help reduce inflammation and support overall health in aging [165,166]. Studies by Cole et al. found that individuals with strong social ties had lower levels of pro-inflammatory gene expression and increased antiviral gene expression compared to socially isolated individuals [167,168].

Smoking cessation is a critical lifestyle intervention for improving immune function. Smoking has been shown to accelerate immunosenescence and increase the risk of various age-related diseases. Quitting smoking can help reduce inflammation and improve immune function, even in older adults who have been long-term smokers [127,169,170]. Studies have shown that within 1 month of quitting smoking, levels of natural killer cells and CD4+ T cells begin to normalize, with further improvements in immune function observed over the following year [171,172,173].

Moderate alcohol consumption is advisable for maintaining immune health. Excessive alcohol intake can impair immune function and increase susceptibility to infections. Limiting alcohol consumption to moderate levels, or abstaining altogether, can help support immune function in aging individuals [174,175,176].

Staying up-to-date with recommended vaccinations is crucial for compensating for the reduced immune response in older adults. Vaccines tailored for older populations, such as high-dose influenza vaccines, have shown improved efficacy in this age group. Regular vaccination can help protect against various infectious diseases and reduce the risk of complications [177,178,179,180].

Weight management is an important aspect of lifestyle interventions for immune health. Obesity is associated with chronic low-grade inflammation, which can accelerate immunosenescence. Maintaining a healthy body weight through a balanced diet and regular physical activity can help reduce inflammation and support immune function [181,182,183].

Implementing these lifestyle interventions as a comprehensive approach can have synergistic effects on immune health and overall well-being in older adults. Henson et al. demonstrated that a multifaceted lifestyle intervention program led to improvements in various markers of immune function in older adults, including increased naive T cell production, enhanced natural killer cell cytotoxicity, and reduced levels of pro-inflammatory cytokines such as IL-6 and TNF-α [184,185].

In conclusion, lifestyle interventions offer a promising approach to mitigating the effects of immunosenescence and promoting healthy aging. By addressing multiple aspects of daily life, including stress management, sleep, social connections, smoking cessation, alcohol consumption, vaccination, weight management, and cognitive stimulation, older adults can take proactive steps to support their immune health and overall well-being. Future research should continue to explore the most effective combinations of lifestyle interventions and their long-term impacts on immune function in aging populations.

Methods:

This comprehensive review was conducted using a systematic approach to literature search, data extraction, and synthesis. The review process involved both traditional research methods and the application of AI-assisted tools to enhance the breadth and depth of our analysis. However, this review article does not contain any original research data, results, images, blots, photographs, X-rays, or measurements that were created, altered, or manipulated using generative artificial intelligence (GenAI) tools.

Literature Search:

We performed an extensive search of electronic databases including PubMed, Web of Science, and Scopus. The search strategy employed a combination of MeSH terms and keywords related to immunosenescence, inflammaging, and lifestyle interventions. The search was limited to peer-reviewed articles published in English within the last 20 years (2007–2024). Additionally, we manually searched the reference lists of relevant review articles to identify any studies that might have been missed in the electronic search.

AI-Assisted Literature Review:

To complement our manual search and to ensure comprehensive coverage of the literature, we utilized Perplexity AI, an advanced language model, to assist in the literature review process. Perplexity AI was employed to perform the following:I.Conduct additional searches using complex query strings to identify relevant articles that might have been missed in the initial manual search.II.Analyze abstracts and full-text articles to extract key information and identify recurring themes.III.Generate summaries of complex scientific concepts to aid in the synthesis of information.

It is important to note that all output generated by Perplexity AI was critically reviewed, extensively edited, and supplemented with additional research and analysis by the human authors. The AI tool was used solely as a supportive tool to enhance the efficiency and comprehensiveness of the review process.

Data Extraction and Synthesis:

Two independent reviewers (AM and LP) extracted data from the selected studies using a standardized form. The extracted data included study characteristics, methodological details, key findings, and conclusions. Any discrepancies in data extraction were resolved through discussion and consensus. The synthesized information was organized into thematic categories aligned with the objectives of the review. These categories included the following:I.Mechanisms of immunosenescence and inflammagingII.Impact on age-related diseasesIII.Lifestyle interventions (diet, exercise, stress management, sleep)IV.Pharmacological approachesV.Emerging therapies

Limitations:

We acknowledge that the use of AI-assisted tools in the review process may have introduced potential biases or limitations. To mitigate this, all AI-generated content was thoroughly reviewed and validated by human experts in the field. Additionally, the search was limited to English language publications, which may have excluded relevant studies published in other languages.

## 6. Conclusions

Immunosenescence and inflammaging represent significant challenges in the aging population, contributing to increased susceptibility to infections, reduced vaccine efficacy, and a higher incidence of age-related diseases. This review has explored the complex molecular and cellular mechanisms underlying these phenomena, including telomere attrition, epigenetic alterations, mitochondrial dysfunction, and changes in key signaling pathways such as NF-κB and mTOR.

Our analysis highlights the intricate interplay between various factors contributing to immunosenescence, including the senescence-associated secretory phenotype (SASP), chronic activation of innate immune sensors, gut microbiome dysbiosis, and metabolic inflammation. Understanding these interconnected processes is crucial for developing targeted interventions to maintain immune function in aging populations.

Emerging research has identified several promising molecular targets for intervention, including the modulation of NF-κB and mTOR pathways, targeting the SASP, activating sirtuins, and regulating AMPK signaling. Additionally, lifestyle interventions such as regular physical activity, a Mediterranean-style diet, stress management, and adequate sleep have shown potential in mitigating aspects of immunosenescence.

Future research should focus on integrating these molecular insights with lifestyle and pharmacological interventions to develop comprehensive strategies for promoting healthy immune aging. This may include combination therapies targeting multiple pathways simultaneously, personalized approaches accounting for individual variations in aging processes, and long-term clinical trials to assess the efficacy and safety of potential interventions.

By advancing our understanding of immunosenescence and developing effective interventions, we can work towards improving healthspan, reducing the burden of age-related diseases, and enhancing the quality of life for older adults. The field of immunosenescence research holds great promise for addressing the challenges of an aging global population and promoting healthy aging for all.

## Figures and Tables

**Table 1 biology-14-00017-t001:** Molecular Targets to prevent immunosenescence and inflammaging.

Molecular Target	Description	Potential Interventions
NF-κB Signaling Pathway	Key regulator of inflammation implicated in age-related diseases	- IKK inhibitors - Nrf2 activators - Natural compounds (e.g., curcumin, resveratrol) [101,102]
mTOR Pathway	Crucial role in cellular metabolism and growth	- Rapamycin and rapalogs - Metformin [103,104,105]
Senescence-Associated Secretory Phenotype (SASP)	Pro-inflammatory secretory phenotype of senescent cells	- Senolytics (e.g., dasatinib + quercetin, fisetin) - SASP inhibitors (e.g., JAK inhibitors, NF-κB inhibitors) [106,107,108]
Sirtuin Pathways	NAD+-dependent deacetylases involved in cellular stress responses	- NAD+ precursors (e.g., NR, NMN) - Sirtuin activators (e.g., resveratrol) [56,57,109,110]
AMPK Signaling	Energy sensor regulating cellular metabolism	- AMPK activators (e.g., AICAR, berberine) - Exercise mimetics [111,112]
Inflammasome Regulation	Central role in innate immunity and age-related inflammation	- NLRP3 inhibitors - IL-1β blockers [113,114]
Epigenetic Modifiers	Age-related changes in epigenetic patterns	- HDAC inhibitors - DNA methyltransferase inhibitors [115,116,117]
Telomerase Activation	Maintenance of telomere length	- Telomerase activators [118,119,120]
Mitochondrial Function	Mitochondrial dysfunction contributes to immunosenescence	- Mitochondrial antioxidants (e.g., MitoQ) - NAD+ boosters [121]
Gut Microbiome Modulation	Crucial role in immune function and inflammation	- Probiotics and prebiotics - Fecal microbiota transplantation [122,123,124]

## Data Availability

No new data were created or analyzed in this study. The article is based on a review of existing published literature. All sources cited in the review are listed in the references section.

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
