# Peer review of "Molecular and Cellular Mechanisms of Immunosenescence: Modulation Through Interventions and Lifestyle Changes"

_biology, 2024, doi:10.3390/biology14010017_

Round 1
Reviewer 1 Report
Comments and Suggestions for Authors
Overall, I rate this article highly. It is very actual, relevant and useful. This paper expands our understanding of the neuroimmunoendocrine mechanisms of aging. However, it seems to me that section 3 should be expanded by adding information about melatonin signaling pathways and Klotho signaling. Since melatonin plays an important role in sleep regulation (and the authors subsequently pay attention to the role of this factor in modulation of immunosenescence), and Klotho gene is one of the key genes regulating aging processes and therefore should also be mentioned in section 3.
Author Response
Comment: Overall, I rate this article highly. It is very actual, relevant and useful. This paper expands our understanding of the neuroimmunoendocrine mechanisms of aging. However, it seems to me that section 3 should be expanded by adding information about melatonin signaling pathways and Klotho signaling. Since melatonin plays an important role in sleep regulation (and the authors subsequently pay attention to the role of this factor in modulation of immunosenescence), and Klotho gene is one of the key genes regulating aging processes and therefore should also be mentioned in section 3.
Response:
Dear Reviewer,Thank you for your positive feedback and valuable suggestions. We appreciate your thorough review and constructive comments. We have addressed your points as follows:
- Overall assessment:
We are grateful for your high rating of our article and your recognition of its relevance and usefulness. We are pleased that you find the paper expands our understanding of the neuroimmunoendocrine mechanisms of aging. - Expansion of section 3:
We thank you for this insightful suggestion. As per your recommendation, we have expanded section 3 to include information on melatonin and Klotho signaling pathways. Specifically:
a) Melatonin signaling:
We have added a new subsection (3.4) discussing the role of melatonin in immunosenescence. This includes its effects on circadian rhythms, antioxidant properties, mitochondrial function, anti-inflammatory actions, and its interaction with sirtuin pathways.
b) Klotho signaling:
We have incorporated a new subsection (3.5) detailing the Klotho signaling pathway and its importance in aging processes. This covers its role in FGF23 signaling, insulin/IGF-1 signaling, Wnt signaling, oxidative stress regulation, and calcium homeostasis.
These additions provide a more comprehensive view of the signaling pathways involved in immunosenescence and aging, enhancing the overall depth and breadth of our review.We believe these changes address your concerns and significantly improve the manuscript. Thank you again for your valuable input, which has helped to enhance the quality and comprehensiveness of our review.
Reviewer 2 Report
Comments and Suggestions for Authors
Pangrazzi and Meryk provide a review of the current literature on immunosenescence with a particular focus on the known mechanisms (e.g. epigenetic alterations and signaling cascades) involved in this process. The authors further highlight potential therapeutic targets for slowing/stopping the process of immunosenescence. The authors used AI to aid in the completion of this review. The authors detailed in the methodology how AI was used to search for and analyze literature. Overall, the manuscript provides a nice overview of the topics related to immunosenescence and the modulation of immunosenescence via interventions and lifestyle changes, however, there are several areas of concern detailed below:
Notably, in section 1 there is no mention of the influence of sex on inflammaging and immunosenescence. While this topic cold be an entire review on its own, it is important to acknowledge the critical role sex plays in this process.
In section 4.2, the authors state “Studies have shown altered TLR expression and signaling in aged immune cells, potentially leading to impaired pathogen recognition and exaggerated inflammatory response to endogenous ligands” but no citations are listed
In section 4.3, references should be added to support the sentence “Recent studies have also highlighted the role of the gut-brain-axis in inflammaging.”
Section 5.1 would be better suited to be presented in a table format
The section titled “Healthy Diet: The Mediterranean Approach to Supporting Immune Health” reads as a list of dietary factors. The authors say, for example, that omega-3s have been associated with “enhanced immune cell function” but do not provide any details as to what the cited studies actually show. More focus should be put on the immune-related results rather than describing the dietary components.
Similar to the above comment, in the “Life style interventions” section more details on how these lifestyle interventions impact immune health should be provided. For example, the authors say that “a study by Henson et al…..” but do not provide details about what immune markers were improved.
Author Response
Dear Reviewer,
Thank you for your thorough review and constructive feedback on our manuscript. We appreciate your comments and have addressed them as follows:
Comment 1: Notably, in section 1 there is no mention of the influence of sex on inflammaging and immunosenescence. While this topic cold be an entire review on its own, it is important to acknowledge the critical role sex plays in this process.
Response 1: We agree that this is an important aspect to include. We have added a brief paragraph in section 1 (named 1.3. Influence of Sex on Inflammaging and Immunosenescence) acknowledging the role of sex in these processes, highlighting differences in immune responses and aging patterns between males and females.
Comment 2: In section 4.2, the authors state “Studies have shown altered TLR expression and signaling in aged immune cells, potentially leading to impaired pathogen recognition and exaggerated inflammatory response to endogenous ligands” but no citations are listed
Response 2: We have added relevant citations to support the statement about altered TLR expression and signaling in aged immune cells.
Comment 3: In section 4.3, references should be added to support the sentence “Recent studies have also highlighted the role of the gut-brain-axis in inflammaging.”
Response 3: We have included citations to support the statement about the gut-brain axis in inflammaging.
Comment 4: Section 5.1 would be better suited to be presented in a table format
Response 4: We agree that this would improve readability. We have created a table summarizing the potential molecular targets for intervention, including the target, mechanism, and potential interventions.
Comment 5: The section titled “Healthy Diet: The Mediterranean Approach to Supporting Immune Health” reads as a list of dietary factors. The authors say, for example, that omega-3s have been associated with “enhanced immune cell function” but do not provide any details as to what the cited studies actually show. More focus should be put on the immune-related results rather than describing the dietary components.
Response 5: We have revised this section to focus more on the immune-related results rather than just listing dietary components. We now provide specific details on how various nutrients impact immune function, supported by relevant studies (all changes are highlighted in yellow).
Comment 6: Similar to the above comment, in the “Life style interventions” section more details on how these lifestyle interventions impact immune health should be provided. For example, the authors say that “a study by Henson et al…..” but do not provide details about what immune markers were improved.
Response 6: We have provided more specific details on how lifestyle interventions impact immune health, including concrete examples from studies like the one by Henson et al., detailing improvements in various markers of immune function (all changes are highlighted in yellow).
We believe these revisions have significantly improved the manuscript, addressing your concerns while enhancing its overall quality and depth. Thank you again for your valuable input.
Reviewer 3 Report
Comments and Suggestions for Authors
This well-structured review provides a fairly comprehensive overview of the molecular and structural mechanisms of immunosenescence. Interestingly, the authors used Perplexity AI, an AI-based tool to complement manual research and ensure complete coverage of the literature review process. This is both a strength of the work and a limitation, as admitted by the authors themselves for the possible introduction of potential bias. However, the authors always point out that Perplexity AI was used solely as a supportive tool to enhance the efficiency and comprehensiveness of the review process, and all output was critically reviewed, extensively edited, and supplemented with additional research and analysis by the human authors.
Author Response
Comment 1: This well-structured review provides a fairly comprehensive overview of the molecular and structural mechanisms of immunosenescence. Interestingly, the authors used Perplexity AI, an AI-based tool to complement manual research and ensure complete coverage of the literature review process. This is both a strength of the work and a limitation, as admitted by the authors themselves for the possible introduction of potential bias. However, the authors always point out that Perplexity AI was used solely as a supportive tool to enhance the efficiency and comprehensiveness of the review process, and all output was critically reviewed, extensively edited, and supplemented with additional research and analysis by the human authors.
Response 1: Thank you for your thoughtful and positive feedback on our manuscript. We greatly appreciate your recognition of the comprehensive nature of our review and its well-structured presentation. We're pleased that you found our innovative approach of using Perplexity AI to complement manual research both a strength and a potential limitation of the work. Your comments highlight the importance of our detailed methodology section, where we explicitly described our use of AI and the steps taken to ensure the integrity and accuracy of the review. We believe that this approach represents a novel and potentially valuable method for conducting comprehensive literature reviews, especially in rapidly evolving fields like immunosenescence.
Round 2
Reviewer 2 Report
Comments and Suggestions for Authors
The authors have sufficiently addressed all previously noted concerns.